# Non-A Blood Type Is a Risk Factor for Poor Cardio-Cerebrovascular Outcomes in Patients Undergoing Dialysis

**DOI:** 10.3390/biomedicines11020592

**Published:** 2023-02-16

**Authors:** Takafumi Nakayama, Junki Yamamoto, Toshikazu Ozeki, Yoshiro Tsuruta, Masashi Yokoi, Tomonori Aoi, Yoshiko Mori, Mayuko Hori, Makoto Tsujita, Yuichi Shirasawa, Chika Kondo, Kaoru Yasuda, Minako Murata, Yuko Kinoshita, Shigeru Suzuki, Michio Fukuda, Chikao Yamazaki, Noriyuki Ikehara, Makoto Sugiura, Toshihiko Goto, Hiroya Hashimoto, Kazuhiro Yajima, Shoichi Maruyama, Kunio Morozumi, Yoshihiro Seo

**Affiliations:** 1Department of Cardiology, Masuko Memorial Hospital, 35–28, Takehashi-cho, Nakamura-ku, Nagoya 453-8566, Aichi, Japan; 2Department of Cardiology, West Medical Center, Nagoya City University, 1-1-1, Hirate-cho, Kita-ku, Nagoya 462-0057, Aichi, Japan; 3Department of Cardiology, Graduate School of Medical Sciences, Nagoya City University, Kawasumi-1, Mizuho-cho, Mizuho-ku, Nagoya 467-0001, Aichi, Japan; 4Division of Nephrology, Graduate School of Medicine, Nagoya University, 65, Tsurumai-cho, Shouwa-ku, Nagoya 466-8550, Aichi, Japan; 5Department of Nephrology, Masuko Memorial Hospital, 35-28, Takehashi-cho, Nakamura-ku, Nagoya 453-8566, Aichi, Japan; 6Clinical Research Management Center, Nagoya City University Hospital, Kawasumi-1, Mizuho-cho, Mizuho-ku, Nagoya 467-0001, Aichi, Japan

**Keywords:** ABO blood type, hemodialysis, cardiovascular event, cerebrovascular event

## Abstract

The clinical impact of ABO blood type on cardio-cerebrovascular outcomes in patients undergoing dialysis has not been clarified. A total of 365 hemodialysis patients participated in the current study. The primary endpoint was defined as a composite including cardio-cerebrovascular events and cardio-cerebrovascular death. The primary endpoint was observed in 73 patients during a median follow-up period of 1182 days, including 16/149 (11%) with blood type A, 22/81 (27%) with blood type B, 26/99 (26%) with blood type O, and 9/36 (25%) with blood type AB. At baseline, no difference was found in the echocardiographic parameters. Multivariable Cox regression analyses revealed that blood type (type A vs. non-A type; hazard ratio (HR): 0.46, 95% confidence interval (95% CI): 0.26–0.81, *p* = 0.007), age (per 10-year increase; HR: 1.47, 95% CI: 1.18–1.84), antiplatelet or anticoagulation therapy (HR: 1.91, 95% CI: 1.07–3.41), LVEF (per 10% increase; HR: 0.78, 95% CI: 0.63–0.96), and LV mass index (per 10 g/m^2^ increase; HR: 1.07, 95% CI: 1.01–1.13) were the independent determinants of the primary endpoint. Kaplan–Meier curves also showed a higher incidence of the primary endpoint in the non-A type than type A (Log-rank *p* = 0.001). Dialysis patients with blood type A developed cardio-cerebrovascular events more frequently than non-A type patients.

## 1. Introduction

An increasing number of patients have been indicated for hemodialysis during the last few decades. According to a survey of the Japanese Society for Dialysis Therapy, 347,671 patients required maintenance dialysis in Japan in 2020, suggesting that the risk stratification of this large cohort is becoming more critical. On the other hand, ABO blood type is a known risk factor for developing several kinds of disease in nondialysis patients, such as infection [1,2,3,4], cancer [3,4,5,6], cognitive disorder [3,7], metabolic disease [3,8], and cardio-cerebrovascular disease [4,9,10,11]. Four main phenotypes of the ABO blood group, i.e., A, B, O, and AB, were differentiated according to the genetically determined antigen presentation on the erythrocytes. ABO antigens are expressed in many organs and widely affect human biology and diseases [3,12]. Although the mechanisms underlying disease vulnerabilities in each ABO blood type are not fully understood, the levels of plasma coagulant factors associated with ABO blood type are considered a crucial substrate for the development of certain kinds of disease [2,3,9,13]. This might directly affect embolic and bleeding diseases, including cardio-cerebrovascular disease.

Patients undergoing dialysis are recognized as being at high risk of both embolic and bleeding diseases. However, the impact of ABO blood type on the clinical outcomes of patients undergoing dialysis has not been clarified. Therefore, in the current study, we aimed to clarify the relationship between ABO blood type and cardio-cerebrovascular outcomes in patients requiring dialysis.

## 2. Materials and Methods

### 2.1. Study Population

All patients requiring dialysis at Masuko Memorial Hospital undergo prescheduled routine echocardiography at least once a year. Reviewing all echocardiographic examinations from April to September 2018, 372 patients were confirmed as undergoing routine echocardiography. After excluding 5 patients who underwent peritoneal dialysis simultaneously with hemodialysis, and 2 patients with RH-blood type, 365 patients were enrolled in the current study (Appendix A). This study was a subanalysis of our previous study, in which we described the clinical impact of left ventricular hypertrophy on patients undergoing dialysis [14]. Because the current study was retrospective and exploratory, the sample size calculation was not performed.

### 2.2. Study Protocol

We retrospectively collected all clinical data from the patients’ medical records. The blood pressure and pulse rate were measured at the start of dialysis on the same day as the last dialysis before the echocardiographic examination. All blood examinations were performed immediately before the daily dialysis. The primary causes of dialysis were categorized as diabetes mellitus and nondiabetes mellitus. A history of cardiovascular disease was defined as ischemic heart disease or open-heart surgery. A history of cerebrovascular disease was defined as cerebral infarction, cerebral hemorrhage, or subarachnoid hemorrhage. Cardiovascular risk factors included hypertension, hyperlipidemia, and diabetes mellitus. Hypertension was defined as a medical history of hypertension or treatment with blood-pressure-lowering therapy; hyperlipidemia was defined as a history of hyperlipidemia, treatment for hyperlipidemia, or a serum low-density lipoprotein level ≥140 mg/dL at baseline; and diabetes mellitus was defined as a history of diabetes mellitus, treatment with glucose-lowering therapy, or a hemoglobin A1c level ≥6.5% at baseline. The primary endpoint was a composite of cardio-cerebrovascular death and admission due to heart failure, a myocardial ischemic event, or a cerebrovascular event. A myocardial ischemic event was defined as any myocardial ischemic event that required revascularization. A cerebrovascular event was defined as cerebral infarction, cerebral hemorrhage, and subarachnoid hemorrhage. Sudden death by an unexplained cause and any death associated with ventricular arrhythmia were included in the definition of cardio-cerebrovascular death. Since most patients were thoroughly followed up through daily dialysis practice at Masuko Memorial Hospital, the information associated with the clinical outcomes could be gathered from medical records. For 14 patients referred to other hospitals before the incidence of the primary endpoint, all information, including referral letters and medical information forms, was reviewed.

The Institutional Ethical Review Board of Masuko Memorial Hospital approved this study (No. MR3-9) and waived the need for informed consent for the study. The information on this retrospective study is available on the Masuko Memorial Hospital website, and all patients were given the opportunity to withdraw from the study. This study was conducted according to the principles of the Helsinki Declaration.

### 2.3. Echocardiographic Measurements

We retrospectively collected the echocardiographic parameters from the routine echocardiography reports, which were performed and assessed following the recommendations of the American Society of Echocardiography [15]. The left ventricular ejection fraction (LVEF) was calculated using the Teichholz method. The left ventricular (LV) wall thickness was the mean intraventricular septal diameter (IVSd) and posterior wall diameter (PWd). The LV mass was calculated according to the following formula [15]: LV mass = 1.04 × ((LVDd + IVSd + PWd)^3^ − LVDd^3^) × 0.8 + 0.6) × 0.001
where LVDd is the LV diastolic diameter. 

The severity of stenotic valvular disease was evaluated at the site of the aortic and mitral valves, as recommended by the American Society of Echocardiography [15]. The grade of regurgitative valvular disease was assessed semiquantitatively by sonographers.

We collected the date interval between the echocardiographic examination and the last dialysis before echocardiography as follows: 0, echocardiography was performed after dialysis on the same day; 1–3, echocardiography was performed 1–3 days after the last dialysis.

### 2.4. Statistical Analysis

Continuous variables are presented as the mean ± standard deviation and were compared using Student’s t-test when normally distributed or presented as the median (interquartile range) and compared using the Mann–Whitney test when not normally distributed. Comparisons of the categorical variables were assessed by Pearson’s chi-squared test. We used Cox proportional hazard analyses to identify the risk factors for the primary endpoint. First, we assessed the prevalence of the primary outcome in each blood type group by the chi-squared test and then divided the patients into two groups: a high-risk blood type group and a non-high-risk blood type group. We summarized the baseline characteristics and compared the parameters according to the two groups. The initial timepoint for the survival analyses was the date of baseline echocardiography (prescheduled routine echocardiography). The day of baseline was also defined as the date of prescheduled routine echocardiography. Univariable Cox proportional hazard analyses were performed using clinical variables that were considered to be risk factors for the primary endpoint in previous scientific reports. Regarding the incidence of the primary endpoint, up to 7 variables that achieved *p*-values < 0.05 in the univariable analyses and were considered the most important for the primary endpoint were entered into the multivariable analyses. When a correlation coefficient >0.70 or <−0.70 was found, or the clinical meaning was almost the same, one of the parameters was excluded from the multivariable Cox regression analyses. The relationships between the echocardiographic parameters and the dialytic date interval were analyzed using Spearman’s test. Kaplan–Meier curves were constructed to compare the event-free survival rate between groups with the log-rank test. Two-sided *p*-values < 0.05 were considered significant. All analyses were performed using SPSS ver.26 (IBM Corp., Armonk, NY, USA). H. Hashimoto reviewed and supervised the statistical analyses in the current study.

## 3. Results

### 3.1. Baseline Characteristics

The primary endpoint was observed in 73 patients during a median follow-up period of 1182 days (interquartile range, 784–1196 days). The incidence of the primary endpoint in each blood type group is presented in Table 1. The primary endpoint was observed significantly less frequently in patients with blood type A (11%, 16/149) than in those with other blood types (non-A type; 26%, 57/216, *p* < 0.001).

The baseline characteristics of all the patients are summarized in Table 2. The pulse rate, history of ischemic heart disease, serum hemoglobin level, and antiplatelet or anticoagulation therapy were significantly different between the two groups (type A and non-A type). None of the echocardiographic parameters were significantly different between the type A and non-A type groups. The LVDd, LV systolic diameter (LVDs), and left atrial diameters were significantly higher according to the dialytic date interval, whereas the LVEF, LV wall thickness, LV mass index, and the ratio of the early diastolic transmitral flow velocity to the mitral annular velocity (E/E′) did not change with the dialytic date interval (Appendix A).

### 3.2. Risk Factors for the Primary Endpoint

Univariable Cox proportional hazard analyses revealed that 10 nonechocardiographic (blood type (type A or not), age, blood pressure, dialytic primary disease (diabetes mellitus or not), history of diabetes mellitus, history of cardio-cerebrovascular disease, serum albumin level, serum sodium level, and antiplatelet or anti-coagulation therapy) and 9 echocardiographic values (LVEF, LVDd, LVDs, LV wall thickness, LV mass index, left atrial diameter, E wave, E/E′, and valvular disease ≥ moderate) were significantly associated with the primary endpoint (Table 3). Blood pressure was excluded because history of hypertension was not a significant parameter. The serum albumin level and serum sodium level were excluded because the mean levels were close between the event-free group and event-occurrence group: serum albumin 3.6 ± 0.4 vs. 3.5 ± 0.3 g/dL and serum sodium 139 ± 3.0 vs. 138 ± 2.8 mEq/L. We selected two echocardiographic parameters, the LVEF and LV mass index, to enter into the multivariable analysis, according to the previous study’s results [14]. For the other parameters that were highly correlated with the primary endpoint, the LVDs and LV wall thickness were excluded because the correlation coefficient between the LVEF and LVDs was -0.82 and the coefficient between the LV mass index and LV wall thickness was 0.75. We did not include the E/E′ in the multivariable analysis because the data were lacking in 74 patients. Finally, multivariable Cox proportional hazard analyses with seven parameters confirmed that blood type (type A vs. non-A type; hazard ratio (HR): 0.46, 95% confidential interval (CI): 0.26–0.81, *p* = 0.007), age (per 10-year increase; HR: 1.47, 95% CI: 1.18–1.84, *p* = 0.001), antiplatelet or anticoagulation therapy (HR: 1.91, 95% CI: 1.07–3.41, *p* = 0.028), LVEF (per 10 % increase; HR: 0.78, 95% CI: 0.63–0.96, *p* = 0.021), and LV mass index (per 10 g/m^2^ increase; HR: 1.07, 95% CI: 1.01–1.13, *p* = 0.020) were independent determinants of the primary endpoint (Table 3). The results did not change if the dialytic date interval was included in the multivariable analysis (Appendix A). The Kaplan–Meier analysis also demonstrated that the primary endpoint was more frequently observed in patients with non-A blood type than those with blood type A (Figure 1, log-rank *p* = 0.001).

### 3.3. Details with each Clinical Outcome and Blood Type

The incidences and comparisons of each clinical outcome of patients with blood type A and non-A-type blood are presented in Table 4, and these are summarized for each blood type in Appendix A. The patients with non-A-type blood had higher incidences of heart failure (*p* = 0.003) and sudden death (*p* = 0.026) than the patients with type A blood (Table 4). The incidence of all-cause death tended to be higher in patients with non-A-type blood than in patients with type A blood (*p* = 0.074), but the difference was not significant. The Kaplan–Meier curves also present a higher rate of sudden death in patients with non-A-type blood (Figure 2A, log-rank *p* = 0.030). The frequency of all-cause death was higher, but not significantly different, in patients with non-A-type blood in the Kaplan–Meier analysis (Figure 2B, log-rank *p* = 0.093).

Dividing the patients into blood types “A or AB” and “B or O” according to the lower and higher incidences of all-cause death resulted in a greater frequency of all-cause death (*p* = 0.038) in patients with blood type A or AB (Appendix A). The causes of death in this study are summarized in Appendix A. The most frequent cause of death was infection, followed by sudden death and heart failure.

## 4. Discussion

We clarified that non-A-type blood (B, O, or AB) is an independent risk factor for developing cardio-cerebrovascular events in patients who require hemodialysis. Our study is the first investigation to assess the clinical impact of ABO blood type on dialysis patients. 

The inclusion of the patients was accompanied by prescheduled routine echocardiography because the current study was an ad hoc analysis of a previous study [14]. However, as the echocardiography findings strongly impact the cardio-cerebrovascular outcomes of patients undergoing dialysis [14], it is appropriate that the initial time point was aligned with the routine echocardiography when assessing the clinical outcomes or comparing the baseline characteristics. The higher frequency of antithrombotic therapy in non-A-type blood patients at baseline was probably explained by the higher rates of history of ischemic heart disease and cerebral infarction. However, our results confirmed that non-A blood type was independent of a history of cardio-cerebrovascular disease for the primary endpoint.

The ABO blood type system consists of the two carbohydrate antigens and antibodies A and B. The blood type is determined genetically by chromosome 9 at locus 9q34. Blood group antigens are expressed widely in not only blood cells, platelets, and leukocytes but also various body secretions, such as seminal fluid, saliva, sweat, urine, gastric secretions, and breast milk [3,12]. According to the Japan Ministry of Health, Labour and Welfare, the distribution of ABO blood type in the Japanese population (2019) was as follows: A, 40.0%; B, 20.0%; O, 30.05%; AB, 9.95%. Our study’s ABO blood type distribution was almost the same (Table 1).

### 4.1. Blood Type A

The ABO blood type antigen is expressed in the adult cardiac endothelium [16,17], and the expression grades seem to be highly individualized [18]. Schussler et al. confirmed that the longevity of cardiac valvular bioprostheses is greater in patients with blood type A than in those with non-A-type blood [19]. Dividing the patients into the two groups of “A or AB” and “B or O” regarding the presence or absence of A antigen, the patients with A or AB had a significantly longer survival after cardiac valve prostheses. We speculate that blood type A plays a protective role in the cardiac endothelium that may prevent the patients from heart failure and sudden death because of the slower progression of valvular heart disease, myocardial inflammation, and thrombosis. 

The consensus that blood type does not influence personality traits is widely accepted [20]. However, based on the fact that the gene for dopamine beta hydroxylase is on chromosome 9q34 and associated with the ABO gene [21,22], some studies have reported possible relationships between ABO blood type and personality traits [23,24]. Tsuchimine et al. demonstrated that blood type AA has a significantly higher persistence score in Japanese subjects than blood types BO and OO, as assessed using the temperament and character inventory score [24]. Persistence is possibly an advantage for the management of maintenance hemodialysis with regard to aspects such as long-term water and diet limitations and continuous attention to the signs of cardio-cerebrovascular events and infection. The mechanisms outlined above are also speculated to affect the lower rates of heart failure and sudden death in patients with blood type A.

### 4.2. Non-A-Type Blood 

The plasma levels of coagulation factor VIII and von Willebrand factor have been reported to be lower in people with blood type O (O < B < A < AB) [3,4,25]. These characteristics are considered to be risk factors for arterial and venous thrombotic events. Conversely, patients with blood type O (and possibly also type B) are also considered at high risk of bleeding. Bayan et al. recognized that blood group O is more frequently observed in patients with upper gastrointestinal bleeding [26]. Furthermore, Takayama et al. demonstrated that blood type O is an independent risk factor for all-cause death in severe trauma patients, possibly due to their lower levels of coagulation factors [27]. In the current study, the higher rate of sudden death with unexplained cause in non-A type patients may be related to bleeding events.

We should be aware of the patient’s ABO blood type when conducting risk stratification for each disease. In the daily practice of dialysis, attending doctors should address symptoms or abnormal laboratory values in the earlier phase for patients with non-A-type blood. In particular, a higher incidence of sudden death must be recognized. Various diseases were reported to be associated with ABO blood type [1,2,3,4,5,6,7,8,9,10,11]; however, none were clarified in dialysis patients. The current study presents the first clinical evidence regarding ABO blood type in dialysis patients by investigating the cardio-cerebrovascular outcomes. Further studies associated with ABO blood type are also warranted, focusing on other diseases in dialysis patients.

### 4.3. Other Factors: Antiplatelet or Anticoagulation Therapy, LVEF, and LV Mass Index

Anticoagulation therapy for patients with atrial fibrillation undergoing dialysis increases the bleeding risk but does not decrease the risk of cerebral infarction [28,29]. Our results support these previous reports. Hence, we should avoid the easy use of antithrombotic therapy in dialytic patients. The independent risk factors did not change after incorporating a history of atrial fibrillation into the multivariable Cox regression analysis (Appendix A).

Reduced LVEF and LV hypertrophy are also well-recognized risk factors for poor clinical outcomes in patients requiring dialysis [14,30,31]. The current study confirmed the importance of transthoracic echocardiography in patients undergoing dialysis. Although significant valvular heart disease was reported to be a risk factor for poor clinical outcomes [14,31], we clarified that our results did not change after including moderate or worse valvular heart disease in the multivariable analysis (Appendix A).

### 4.4. Study Limitations

The current study had several limitations. This study was retrospective and observational in design and carried out at a single center. Since the current study was conducted as exploratory, sample size calculation was not performed. As the distribution of blood types varies across countries and races, the results cannot be directly applied to non-Japanese subjects. We could not differentiate the detailed genetic ABO phenotype, such as AA or AO and BB or BO. The incidence of sudden death by unexplained cause was relatively high, meaning that in the current study, we could not sufficiently discuss the cause of death. The current study only described the statistical associations between ABO blood type and clinical events in dialysis patients, not their underlying mechanisms. A prospective, long-term follow-up, multicenter study with detailed ABO genotype information is required to confirm our results.

## 5. Conclusions

Patients with blood type A who required dialysis had a significantly lower incidence of cardio-cerebrovascular events than those with other blood types. Heart failure and sudden death were more frequently observed in blood types B, O, and AB compared to A. We should be mindful of cardio-cerebrovascular events in these non-A blood types in daily hemodialytic practice.

## Figures and Tables

**Figure 1 biomedicines-11-00592-f001:**
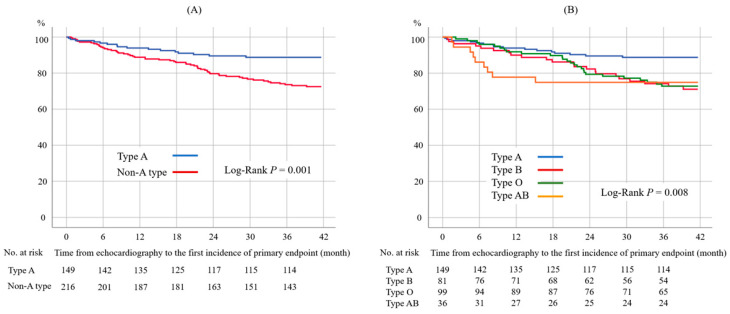
Kaplan–Meier curves for the primary endpoint according to type A or non-A type (**A**) and each blood type (**B**). Patients with dialysis and blood type A had a significantly lower incidence of the primary endpoint compared to those with dialysis and non-A type.

**Figure 2 biomedicines-11-00592-f002:**
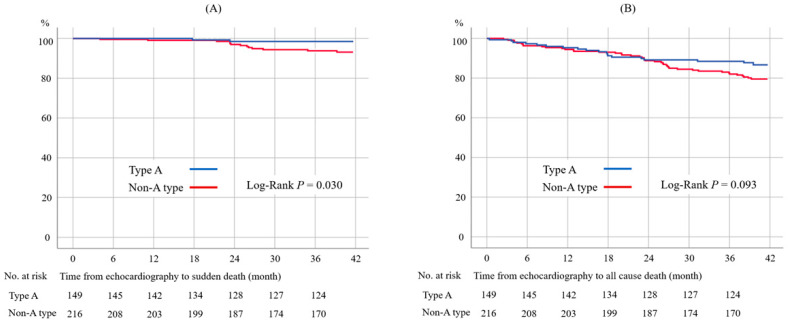
Kaplan–Meier survival curves for sudden death by unexplained cause (**A**) and all-cause death (**B**). The incidence of sudden death was significantly lower in patients with blood type A than non-A type (**A**). The rate of all-cause death tended to be lower in blood type A than non-A type but was not statistically significant (**B**).

**Table 1 biomedicines-11-00592-t001:** The incidence of the primary endpoint and each event in each blood type.

ABO Blood Type	A	B	O	AB	Total
The number of patients, n (%)	149 (41%)	81 (22%)	99 (27%)	36 (10%)	365
Primary endpoint	16 (11%)	22 (27%)	26 (26%)	9 (25%)	73

The primary endpoint was more frequently observed in patients with blood type A than patients with non-A type blood by chi-square test.

**Table 2 biomedicines-11-00592-t002:** Baseline characteristics.

Characteristic	All Patients	Type A	Non-A Type	*p*-Value
	*n* = 365	*n* = 149	*n* = 216	
Basic data				
Age, y	65.7 ± 12.5	64.4 ± 13.0	66.6 ± 12.1	0.10
Male	252 (69%)	107 (72%)	145 (67%)	0.37
BMI, kg/m^2^	22.1 ± 4.6	22.2 ± 4.5	22.0 ± 4.7	0.78
Blood pressure, mmHg	146.8 ± 23.4	146.8 ± 25.2	146.9 ± 22.1	0.98
Pulse rate, bpm	72.9 ± 11.6	74.5 ± 11.7	71.8 ± 11.4	0.026
Dialysis duration, y	9.8 ± 8.7	9.7 ± 8.4	9.9 ± 8.9	0.82
Dialytic date interval, d				0.67
0	49 (13%)	19 (13%)	30 (14%)	
1	190 (52%)	83 (56%)	107 (50%)	
2	78 (21%)	28 (19%)	50 (23%)	
3	48 (13%)	19 (13%)	29 (13%)	
Primary disease of dialysis				
Diabetes mellitus	120 (%)	47 (32%)	73 (34%)	0.65
Cardiovascular risk factors				
Hypertension	259 (71%)	103 (69%)	156 (72%)	0.52
Dyslipidemia	127 (35%)	51 (34%)	76 (35%)	0.85
Diabetes mellitus	143 (39%)	59 (40%)	84 (39%)	0.89
History of atrial fibrillation	21 (6%)	6 (4%)	15 (7%)	0.24
History of cardiovascular disease				
Ischemic heart disease	65 (18%)	18 (11%)	47 (22%)	0.018
Cardiovascular surgery	21 (6%)	10 (7%)	11 (5%)	0.51
History of cerebrovascular disease				
Cerebral infarction	24 (7%)	7 (5%)	17 (8%)	0.23
Cerebral hemorrhage	6 (2%)	1 (1%)	5 (2%)	0.23
Subarachnoid hemorrhage	2 (1%)	1 (1%)	1 (0%)	0.79
Laboratory measurements				
Hemoglobin, g/dL	11.2 ± 1.1	11.3 ± 1.2	11.1 ± 1.1	0.040
Platelets, ×10^4^/µg	18.6 ± 6.1	18.8 ± 6.3	18.5 ± 5.9	0.64
Albumin, g/dL	3.6 ± 0.4	3.6 ± 0.4	3.6 ± 0.4	0.84
Total bilirubin, mg/dL	0.3 (0.2–0.4)	0.3 (0.2–0.4)	0.3 (0.2–0.4)	0.79
AST, IU/L	13.1 ± 6.2	13.2 ± 6.8	13.0 ± 6.0	0.82
ALT, IU/L	10.5 ± 5.8	10.7 ± 6.2	10.2 ± 5.6	0.42
LDH, IU/L	183.7 ± 35.4	185.8 ± 40.0	182.2 ± 31.9	0.34
BUN, mg/dL	57.6 ± 14.5	56.1 ± 15.4	58.6 ± 13.8	0.11
Uric acid, mg/dL	7.1 ± 1.6	7.2 ± 1.5	7.0 ± 1.6	0.22
Serum sodium, mEq/L	138.8 ± 3.0	138.8 ± 3.5	138.8 ± 2.6	0.56
Serum phosphorus, mmol/L	5.1 (4.3–5.8)	5.1 (4.4–5.8)	5.1 (4.3–5.9)	0.85
Serum iron, µg/dL	55 (39–64)	50 (40–64)	50 (38–65)	0.96
Medication				
B-blocker	132 (36%)	54 (36%)	78 (36%)	0.98
ACEI/ARB	153 (42%)	60 (40%)	93 (43%)	0.60
Ca-blocker	190 (52%)	82 (55%)	108 (50%)	0.34
Statin	95 (26%)	38 (26%)	57 (26%)	0.85
Antiplatelet or anticoagulation	133 (36%)	45 (30%)	88 (41%)	0.037
Echocardiography				
LVEF, %	65.3 ± 10.7	65.5 ± 10.5	65.1 ± 10.8	0.71
LV diastolic diameter, mm	47.5 ± 6.4	47.1 ± 6.4	47.7 ± 6.4	0.36
LV systolic diameter, mm	30.3 ± 6.4	30.0 ± 6.2	30.6 ± 6.6	0.39
LV wall thickness, mm	10.3 ± 1.9	10.2 ± 1.7	10.4 ± 2.0	0.34
LV mass index, g/m^2^	110.5 ± 34.4	106.6 ± 33.8	113.3 ± 34.7	0.069
Left atrial diameter, mm	36.3 ± 6.2	36.5 ± 6.0	36.2 ± 6.3	0.64
E wave, cm/s	75.3 ± 24.2	74.3 ± 24.3	76.0 ± 24.2	0.50
Deceleration time, ms	227.3 ± 66.5	226.4 ± 61.2	228.0 ± 70.0	0.82
E/E′	11.5 ± 5.0	11.0 ± 4.8	11.8 ± 5.1	0.20
Valvular disease ≥ moderate	46 (13%)	14 (9%)	32 (15%)	0.13

Values are the number (%) or mean ± SD, as indicated. Abbreviations: BMI, body mass index; AST, aspartate aminotransferase; ALT, alanine aminotransferase; LDH, lactate dehydrogenase; BUN, blood urea nitrogen; ACEI, angiotensin-converting enzyme inhibitor; ARB, angiotensin 2 receptor blocker; LV, left ventricular; EF, ejection fraction; E wave, the early diastolic transmitral flow velocity; E/E′, ratio of the early diastolic transmitral flow velocity to mitral annular velocity.

**Table 3 biomedicines-11-00592-t003:** Cox proportional hazard analyses.

Characteristic	Univariable Analyses			Multivariable Analyses		
	HR	95% CI	*p*-Value	HR	95% CI	*p*-Value
Blood type						
Type A, vs. non-A type	0.39	0.23–0.69	0.001	0.46	0.26–0.81	0.007
Basic data						
Age, per 10-year increase	1.54	1.25–1.90	<0.001	1.47	1.18–1.84	0.001
Male, vs female	0.97	0.61–1.66	0.97			
BMI, kg/m^2^	1.03	0.99–1.08	0.17			
Blood pressure, per 10 mmHg increase	1.12	1.02–1.24	0.020	-	-	-
Pulse rate, per 10 bpm increase	1.11	0.91–1.35	0.32			
Dialysis duration, y	0.99	0.97–1.02	0.66			
Primary disease of dialysis						
Diabetes mellitus, vs. nondiabetes mellitus	1.63	1.04–2.56	0.033	1.24	0.76–2.02	0.39
Cardiovascular risk factors						
Hypertension	0.92	0.56–1.52	0.75			
Dyslipidemia	1.00	0.62–1.61	1.00			
Diabetes mellitus	1.60	1.01–2.53	0.045	-	-	-
History of atrial fibrillation	2.58	1.24–5.37	0.012	-	-	-
History of cardio- or cerebrovascular disease	2.75	1.73–4.37	<0.001	1.14	0.63–2.07	0.66
Laboratory measurements						
Hemoglobin, g/dL	0.95	0.77–1.17	0.64			
Platelets, ×10^4^/µg	0.99	0.95–1.03	0.51			
Albumin, g/dL	0.42	0.24–0.75	0.004	-	-	-
Total bilirubin, mg/dL	1.12	0.57–2.23	0.74			
AST, IU/L	1.00	0.97–1.04	0.88			
ALT, IU/L	0.98	0.94–1.02	0.37			
LDH, IU/L	1.01	0.50–1.08	0.75			
BUN, mg/dL	1.00	0.98–1.02	0.86			
Uric acid, mg/dL	0.94	0.81–1.09	0.41			
Serum sodium, mEq/L	0.91	0.85–0.97	0.004	-	-	-
Serum calcium, mg/dL	1.04	0.75–1.45	0.83			
Serum phosphorus, mmol/L	0.99	0.91–1.07	0.71			
Serum ferritin, per 10 ng/mL increase	1.01	1.00–1.03	0.19			
Serum iron, µg/dL	0.99	0.98–1.00	0.20			
Serum Int-PTH, per 10 pg/mL increase	1.00	0.98–1.02	0.67			
Medication						
B-blocker	1.42	0.90–2.26	0.14			
ACEI/ARB	1.03	0.65–1.64	0.89			
Antiplatelet or anticoagulation	3.01	1.88–4.82	<0.001	1.91	1.07–3.41	0.028
Echocardiography						
LVEF, per 10% increase	0.67	0.55–0.82	<0.001	0.78	0.63–0.96	0.021
LV diastolic diameter, mm	1.04	1.00–1.08	0.041	-	-	-
LV systolic diameter, mm	1.06	1.03–1.10	<0.001	-	-	-
LV wall thickness, mm	1.20	1.09–1.33	<0.001	-	-	-
LV mass index, per 10 g/m^2^ increase	1.13	1.07–1.20	<0.001	1.07	1.01–1.13	0.020
Left atrial diameter, mm	1.05	1.01–1.09	0.014	-	-	-
E wave, per 10 cm/s increase	1.12	1.03–1.22	0.010	-	-	-
Deceleration time, per 10 ms increase	0.98	0.95–1.02	0.32			
E/E′ (*n* = 290)	1.10	1.06–1.14	<0.001	-	-	-
Valvular disease ≥ moderate	2.92	1.71–4.98	<0.001	-	-	-

**Table 4 biomedicines-11-00592-t004:** The incidence of each clinical outcome in patients with type A and non-A type.

ABO Blood Type	Type A	Non-A Type	*p*-Value
The Number of Patients, n	149	216	
Heart failure, n (%)	3 (2%)	21 (10%)	0.003
Ischemic heart event, n (%)	6 (4%)	10 (5%)	0.78
Cerebrovascular event, n (%)	5 (3%)	17 (8%)	0.075
Sudden death, n (%)	2 (1%)	13 (6%)	0.026
All cause death, n (%)	19 (13%)	43 (20%)	0.074

## Data Availability

The datasets used and/or analyzed during the current study are available from the corresponding author on reasonable request. Correspondence and requests for materials should be addressed to T.N.

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
