# Peer review of "Non-A Blood Type Is a Risk Factor for Poor Cardio-Cerebrovascular Outcomes in Patients Undergoing Dialysis"

_biomedicines, 2023, doi:10.3390/biomedicines11020592_

Round 1

Reviewer 1 Report

Dear Authors,

you presented an interesting study ! Concerning blood-types most studies reveal an opposite result - that type A individuals are at high risk of cardiovascular events, especially in Caucasians. However, the association is not clear. You reveal a different association in the hemodialysis patients  - however, you discuss the limitations of the study.

I would like (if you have the data) to explain the indication of antithrombotic treatment in non-blood A patients - why was more frequent?

Reviewer 2 Report

The authors present an interesting manuscript, well focused and with interesting data. However, you need to make changes:

-The authors should change the title and be more direct.

-The authors must improve the summary, providing precise data and giving a practical character to the results.

-The authors should improve the references included, they should be more current.

-The authors must justify the innovation of this study compared to others in the field of biomedicine.

-The authors should substantially improve the way they present the results. The quality of the figures is poor.

-Authors must adequately describe the results.

-Authors must include representative images of the patients.

-The sample size must be statistically justified, calculating the statistical power.

-The authors should carry out a more practical discussion, from the clinical point of view.

-Authors should extensively improve their use of English grammar.

Reviewer 3 Report

biomedicines-2211334_ Impact of ABO Blood Type on Cardio-cerebrovascular Outcomes in Patients Undergoing Dialysis

The aim of this study is to clarify the relationship between ABO blood type and cardio-cerebrovascular outcomes in patients requiring dialysis.

The introduction should broadly state the hypothesis that gives rise to this work (such as: There are four main phenotypes of ABO blood groups i.e. A, B, O and AB were differentiated based on the type of antigen present on the erythrocytes…..)

In material and methods, the follow-up that has been carried out on the patients should be written, as well as the design used. Ethics committee approval must also be submitted. What inclusion and exclusion criteria were used for the patients in this study? Was a sample size calculation performed?

Table 1 shows the baseline characteristics of the patients. Do these characteristics correspond to the time they were included in the follow-up?

Whereas a few studies have been carried out on the association of ABO blood groups with cardiovascular diseases but there is not in dialysis, this article could support light in this topic.

Round 2

Reviewer 2 Report

The authors have adequately answered the reviewer's questions. However, the authors must include the lack of images in the limitations of the study. The figures are still of low quality.

Author Response

Reply to Reviewer #2

The authors would like to thank the reviewer for the whole time to review the manuscript.

Reviewer’s comment_1)

However, the authors must include the lack of images in the limitations of the study.

Reply)

The authors are sorry that the reply might not be sufficient to express the intention. The current study did not investigate something requiring images, so the current study does not have any images to represent. Not lacking images that should be represented.

Reviewer’s comment_2)

The figures are still of low quality.

Reply)

The authors understand the importance of well-designed and well-visualized figures. However, as the reviewer has read, the current study had only Kaplan-Meyer curves to present the results as figures. The authors appreciate it very much if the reviewer indicates specific suggestions regarding the quality of the figures as a scientific research paper.

Reviewer 3 Report

biomedicines-2211334_ Impact of ABO Blood Type on Cardio-cerebrovascular Outcomes in Patients Undergoing Dialysis

I have carefully reviewed the response of the authors, as well as the new version of the manuscript in which I see the suggestion of the other reviewers incorporated. I consider that the article has improved in clarity and can better present its results. However, they not calculating the sample size implies a weakness or limitation that should be in the limitations part of the study.

Author Response

Reply to Reviewer #3

The authors would like to appreciate the time and careful review of our manuscript.

Reviewer’s comment_1)

However, they not calculating the sample size implies a weakness or limitation that should be in the limitations part of the study.

Reply)

The authors got to understand the reviewer’s valuable comments. The authors added the limitation regarding not calculating the sample size.

<Page 21, Lines 333-334>

Before revised: none

After revised:

Since the current study was conducted exploratory, sample size calculation was not performed.